∂ | **Open Peer Review** | Clinical Microbiology | Research Article

# Distinct compositions and functions of circulating microbial DNA in the peripheral blood compared to fecal microbial DNA in healthy individuals

Taiyu Zhai,[1] Wenbo Ren,[1] Xufeng Ji,[1] Yifei Wang,[2] Haizhen Chen,[1] Yuting Jin,[1] Qiao Liang,[1] Nan Zhang,[1] Jing Huang[1]

**ABSTRACT**  The crucial function of circulating microbial DNA (cmDNA) in peripheral blood is gaining recognition because of its importance in normal physiology and immunity in healthy individuals. Evidence suggests that cmDNA in peripheral blood is derived from highly abundant, translocating gut microbes. However, the associations with and differences between cmDNA in peripheral blood and the gut microbiome remain unclear. We collected blood, urine, and fecal samples from volunteers to compare their microbial information via 16S rDNA sequencing. The results revealed that, compared with gut microbial DNA, cmDNA in peripheral blood was associated with reduced diversity and a distinct microbiota composition. The cmDNA in the blood reflects the biochemical processes of microorganisms, including synthesis, energy conversion, degradation, and adaptability, surpassing that of fecal samples. Interestingly, cmDNA in blood showed a limited presence of DNA from anaerobes and gram-positive bacteria, which contrast with the trend observed in fecal samples. Furthermore, analysis of cmDNA revealed traits associated with mobile elements and potential pathologies, among others, which were minimal in stool samples. Notably, cmDNA analysis indicated similarities between the microbial functions and phenotypes in blood and urine samples, although greater diversity was observed in urine samples. Source Tracker analysis suggests that gut microbes might not be the main source of blood cmDNA, or a selective mechanism allows only certain microbial DNA into the bloodstream. In conclusion, our study highlights the composition and potential functions associated with cmDNA in peripheral blood, emphasizing its selective presence; however, further research is required to elucidate the mechanisms involved.

**IMPORTANCE**  Our research provides novel insights into the unique characteristics and potential functional implications of circulating microbial DNA (cmDNA) in peripheral blood. Unlike other studies that analyzed sequencing data from fecal or blood microbiota in different study cohorts, our comparative analysis of cmDNA from blood, urine, and fecal samples from the same group of volunteers revealed a distinct blood-specific cmDNA composition. We discovered a decreased diversity of microbial DNA in blood samples compared to fecal samples as well as an increased presence of biochemical processes microbial DNA in blood. Notably, we add to the existing knowledge by documenting a reduced abundance of anaerobes and gram-positive bacteria in blood compared to fecal samples according to the analysis of cmDNA and gut microbial DNA, respectively. This observation suggested that a potential selective barrier or screening mechanism might filter microbial DNA molecules, indicating potential selectivity in the translocation process which contrasts with the traditional view that cmDNA primarily originates from random translocation from the gut and other regions. By highlighting these differences, our findings prompt a reconsideration of the origin and role of cmDNA

Address correspondence to Jing Huang, huangj@jlu.edu.cn.

Taiyu Zhai and Wenbo Ren contributed equally to this article. Author order was determined through discussion among all the contributing authors.

The authors declare no conflict of interest.

See the funding table on p. 15.

in blood circulation and suggest that selective processes involving more complex biological mechanisms may be involved.

**KEYWORDS** circulating microbial DNA, peripheral blood, gut microbiota, origin, composition

The close connection between microorganisms and diseases is widely recognized (1). At present, studies exploring this relationship have focused on the gut microbiome, as assessed in stool samples. Nevertheless, there are distinct advantages to using blood samples in terms of ease of storage and minimization of bacterial contamination. Examining the genetic material of microorganisms present in blood holds promise for the investigation of disease mechanisms and the development of novel diagnostic approaches.

Blood is critical in all bodily functions, and many physiological or pathological changes can lead to modifications in its composition. For instance, a high-fat diet or diabetes can crucially alter lipid and glucose levels in the bloodstream, and iron-rich foods can increase the production of red blood cells, while a diet lacking vitamin B12 or folic acid can lead to anemia (2). In addition, the latest evidence indicates that changes in blood composition include changes in the genetic material of various types of microorganisms (3). Studies have confirmed the presence of circulating microbial DNA (cmDNA) in the human bloodstream (4). Moreover, recent reports have revealed notable discrepancies in both the concentration and composition of cmDNA in the peripheral blood of patients compared to healthy individuals. Alterations in the features of peripheral blood cmDNA seem to correlate with the progression of specific ailments, such as diabetes and cardiovascular diseases (5, 6). Moreover, monitoring these changes is important for assessing cancer prognosis, identifying patients with Parkinson's disease, and predicting the risk of sepsis in children (7–9).

Despite the vast potential of cmDNA in the diagnosis and treatment of diseases, several aspects, such as its origin, remain poorly understood. Researchers have suggested that the presence of microbial DNA in peripheral blood is merely coincidental, and this DNA is not directly involved in the development of disease; rather, it represents only an "innocent bystander" or "transient visitor" that accidentally enters the peripheral blood, as there is a lack of direct evidence demonstrating a causal relationship between the absence of cmDNA and the onset of disease (10). However, new evidence suggests that blood microorganisms in healthy humans are critical in maintaining normal physiology and immunity, and changes in cmDNA in peripheral blood may represent alterations in human pathophysiology (11). Additionally, bacterial DNA can activate the host immune system, leading to complex pathological pathways, such as those associated with metabolic dysfunction and endothelial damage, contributing to the development of cardiometabolic diseases (12). This evidence suggests that the presence of cmDNA in blood is not incidental. Instead, these findings may have important physiological and pathological significance.

The origin of cmDNA in blood is a topic of great interest. In patients with trauma or infectious diseases, microorganisms in specific regions may enter the peripheral bloodstream, resulting in temporary bacteremia or sepsis. These conditions may alter the composition and quantity of cmDNA in peripheral blood (12). In addition, in healthy individuals or patients with noncommunicable diseases, it is widely believed that the blood microbiota is generated mainly via the translocation of microorganisms from other microbe-rich sites, such as the gut, oral cavity, respiratory tract, and urogenital system (3). Tan et al. suggested the use of the term "cmDNA" instead of "blood microbiota" because evidence supports the absence of a core microbiome in the bloodstream; however, this topic requires further exploration and verification (13). Regardless of the terminology used, given that the gut is the largest site of microbial colonization in the human body and that microbes can translocate from the gut into the peripheral blood, the

connections and differences between cmDNA detected in the blood and gut microbial DNA warrant in-depth study.

A study has shown that the composition of bacteria in the blood of healthy individuals differs from the composition of bacteria in the intestines. The most common intestinal microbiota phyla are *Firmicutes* and *Bacteroidota*, whereas *Proteobacteria* has been reported to be the predominant phylum in the blood microbiota (3). Despite these findings, the researchers compared only microbial diversity in fecal and blood samples. Importantly, due to the potential influences of dietary habits and the living environment on the gut microbiota, the use of data from different research cohorts may not have enabled comprehensive elucidation of the connection between blood cmDNA and gut microbial DNA (14). To date, no further evidence has been provided to support the differences or associations between cmDNA in blood and gut microbial DNA, and understanding this aspect is crucial for determining the origin of cmDNA.

In this study, our primary objective was to conduct a thorough exploration and comparison of the differences between cmDNA present in peripheral blood and microbial DNA in fecal samples. Understanding these disparities may illuminate the unique characteristics and origins of cmDNA within the human body. To ensure reliable and consistent results, we employed a rigorous sampling protocol. We collected blood, fecal, and urine samples concurrently from a carefully selected cohort of healthy volunteers. This approach was designed to minimize the confounding effects of variable dietary and environmental factors on our sequencing data. By doing so, we aimed to unravel the source and subsequent processes influencing the presence of cmDNA in the human bloodstream. We believe that the insights gained from these analyses will enhance our understanding of cmDNA and potentially establish its relevance in clinical settings, particularly in the areas of diagnosis and disease management.

## RESULTS

### The microbial diversity in peripheral blood indicated by cmDNA analysis is significantly reduced

Peripheral blood was collected from volunteers, the blood was cultured, and no replication-competent microorganisms were found. The physical examination data for the 26 volunteers participating in this study are summarized in Table 1. For the purposes of this research, these volunteers were considered healthy because their biochemical parameters fell within the normal range and because they had no symptoms of discomfort. It is crucial to acknowledge, however, that additional medical evaluations were not conducted to assess overall health status. Among the samples from the 26 patients, 610 cmDNA amplicon sequence variants (ASVs) were detected in blood

TABLE 1 Basic information of the 26 healthy volunteers[a]

| Characteristics (mean) | Males (*n* = 12) | Females (*n* = 14) | Normal reference ranges |
|---|---|---|---|
| Age | 32 (27–37) | 32 (21–42) | NA |
| BMI (SD) | 21.75 ± 3.13 | 19.55 ± 2.92 | NA |
| LDL (mmol/L) | 2.62 ± 0.46 | 2.68 ± 0.44 | ≤3.12 |
| HDL (mmol/L) | 1.44 ± 0.10 | 1.45 ± 0.11 | 1.04–1.55 |
| TG (mmol/L) | 1.27 ± 0.36 | 1.24 ± 0.32 | 0.56–1.7 |
| TC (mmol/L) | 4.46 ± 0.66 | 4.33 ± 0.69 | 2.8–5.2 |
| RBC (×10$^{12}$/L) | 4.73 ± 0.44 | 4.26 ± 0.38 | M4.0–5.5/F3.5–5.0 |
| WBC (×10$^{9}$/L) | 6.88 ± 1.52 | 6.17 ± 1.37 | 4.0–10.0 |
| U-Ket | – | — | Negative |
| U-Pro | – | — | Negative |
| BP (mmHg) | 126/83 ± 11.6/7.9 | 117/76 ± 9.8/7.5 | 90–139/60–89 |
| FBG (mmol/L) | 4.65 ± 1.48 | 4.25 ± 1.44 | 3.9–6.1 |

[a]BMI: Body mass index. LDL: Low density lipoprotein. HDL: High density lipoprotein. TG: Triglyceride. TC: Total cholesterol. RBC: Red blood cell. WBC: White blood cell. U-Ket: Urine-ketones. U-Pro: Urine-protein. BP: Blood pressure. FBG: Fasting blood glucose.

samples, 1,685 in fecal samples, and 1,657 in urine samples (Fig. 1A). Alpha diversity analysis revealed significantly lower [Kruskal−Wallis test, false discovery rate (FDR)-adj. $P \leq 0.05$] Chao1 and Shannon indices (Fig. 1B and C) and a significantly greater (Kruskal−Wallis test, FDR-adj. $P \leq 0.05$) Good's coverage index (Fig. 1D) in the blood samples than in the urine and fecal samples. The Pielou-E index of blood was significantly lower (Kruskal−Wallis test, FDR-adj. $P \leq 0.05$) than in the feces, but the difference was not significant (Kruskal−Wallis test, FDR-adj. $P > 0.05$) between the blood and urine

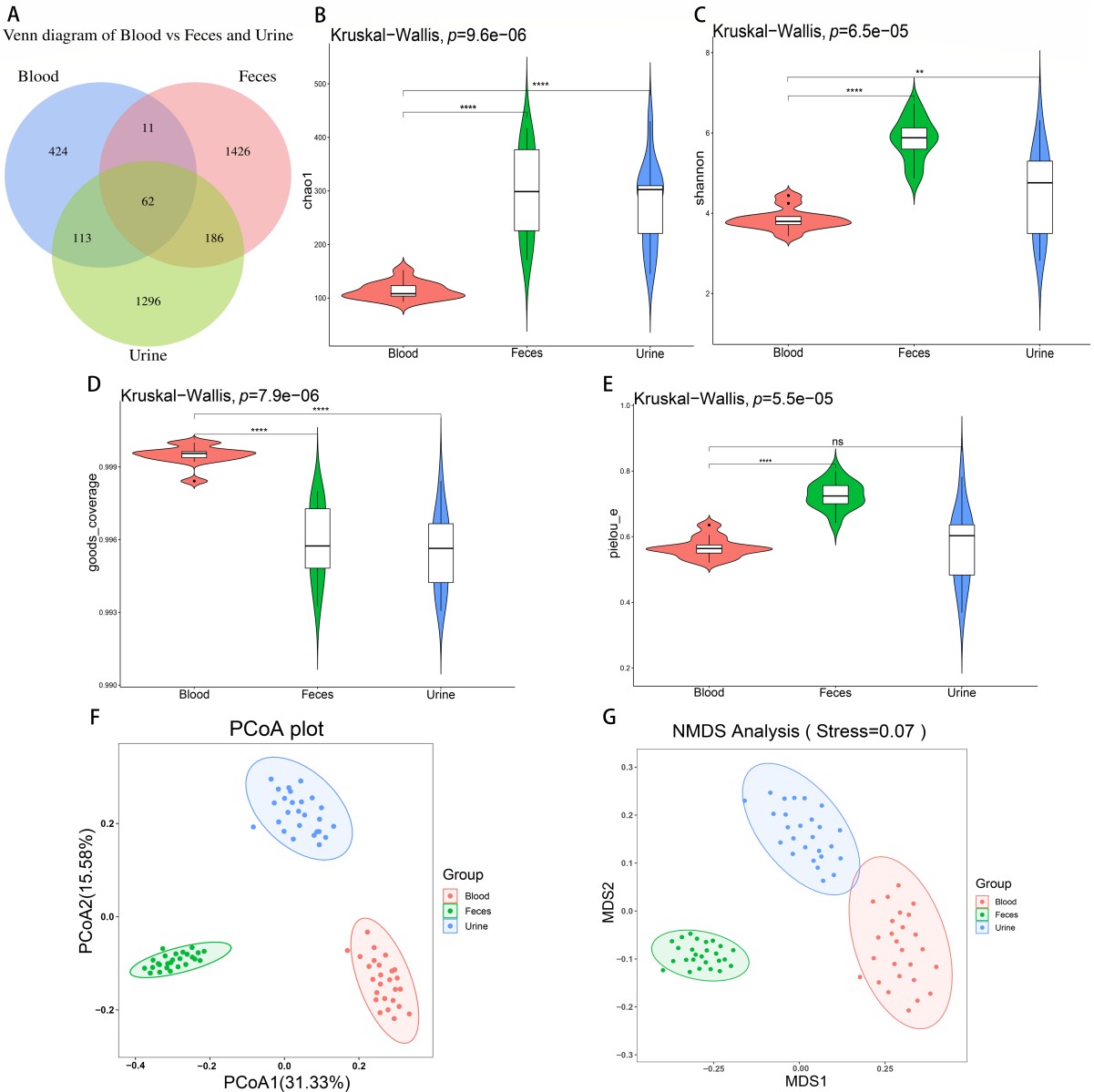

FIG 1 The relative ASV abundance and species diversity of the blood cmDNA samples were significantly different from those of the fecal or urine samples. (A) The Venn diagram displays group overlaps, with circles representing ASV commonality and uniqueness. (B) Chao1 index analysis revealed that the total number of ASVs in blood was lower than that in feces and urine. (C) The Shannon index suggested the presence of unidentified community components, with blood showing lower diversity than feces and urine. (D) Goods coverage indicates the representativeness of the sequencing data relative to the actual samples, and the results indicate comprehensive microbial coverage. (E) The Pielou-E index, similar to Shannon's evenness, showed lower evenness in blood samples than in feces, with no significant difference compared to urine. (F) PCoA revealed notable disparities in microbial community structure between groups based on unweighted UniFrac distances at the phylum and genus levels. (G) NMDS analysis revealed distinct variations in microbial community structure among the groups based on unweighted UniFrac distances. * indicates a significant difference, ** indicates a highly significant difference, and ns signifies no significant difference.

samples (Fig. 1E). Samples from each group were subsequently processed for beta diversity analysis based on unweighted UniFrac distances. According to the principal coordinate analysis (PCoA), the microbial composition differences between the plasma and urine samples were minimal, indicating a high degree of similarity. On the other hand, the differences in microbial community structure between the plasma and fecal samples were more substantial than those between the other samples, indicating greater dissimilarity (Fig. 1F). Similar results were obtained by non-metric multidimensional scaling (NMDS) analysis, and the stress value was 0.07 (Fig. 1G). Analysis of similarity (ANOSIM) revealed significant differences (Permutation test, $N = 78$; FDR-adj. $P \leq 0.05$) in species composition between groups as well as a few differences within groups, with an $R$ value of 0.985246.

## The species in blood samples are significantly different from those in fecal and urine samples

For a more accurate analysis of species composition, we used Silva as well as the NT-16S database for species classification and subsequent analysis, with an annotation threshold of confidence greater than 0.7. Based on the ASV annotation results and the ASV abundance table of each sample, we analyzed the species abundances of samples from different groups at the phylum and genus taxonomic levels. *Proteobacteria* was the core phylum in the blood samples, while *Firmicutes* and *Bacteroidota* were the predominant bacteria in the fecal samples. Unlike in the blood samples, in the urine samples, *Proteobacteria*, *Firmicutes*, and *Bacteroidota* were more abundant, while *Actinobacteriota* was also a dominant group (Fig. 2A). At the genus taxonomic level, *Herminiimonas* exhibited an elevated abundance in the blood samples compared to the other samples, while *Bacteroides* and *Faecalibacterium* were dominant in the fecal samples. In the urine samples, *Lactobacillus* and *Gardnerella* were the primary genera detected (Fig. 2B). Based on the relative abundance table of ASVs at each level, we conducted a differential analysis of all the species, screened the top 30 species with $P$ values less than 0.05 and constructed a histogram. The results in Fig. 2C and D further demonstrate the differences in the abundance of various categories of microorganisms at the phylum and genus levels in the three sample groups. We made detailed comparison data available in Tables S1 and S2. Figure 2C and D present the meaningful comparison groups (Kruskal–Wallis test, FDR-adj. $P \leq 0.05$), while the groups without statistical significance are not included in the figures (Kruskal–Wallis test, FDR-adj. $P > 0.05$).

## Blood cmDNA is associated with a unique dominant species

Figure 3A shows the relative abundance and intergroup differences of the 30 most common species in the different samples. Among the 30 microbes with the most prominent abundance in the fecal and urine samples, six were not detected in the blood samples, namely, *Prevotella*, *Lachnoclostridium*, *Agathobacter*, *Roseburia*, *Prevotella-9*, and *Subdoligranulum* (Fig. 3A). We subsequently analyzed the proportion of the dominant species in each group and the distribution of each dominant species among the different groups. At the phylum level, *Proteobacteria* was the dominant taxon in the blood, accounting for more than 93% of the bacteria. The dominant taxa in the feces were *Firmicutes* (55%) and *Bacteroidota* (28%). In the urine samples, the dominant phyla were *Proteobacteria* and *Firmicutes*, with proportions of 48% and 25%, respectively (Fig. 3B). At the genus level, the dominant taxon in the blood samples was *Herminiimonas*, with a proportion greater than 72%. In the fecal samples, *Bacteroides* was the dominant taxon, with a proportion greater than 99%, while in the urine samples, the dominant taxon was *Lactobacillus*, with a proportion of approximately 38%, followed by *Herminiimonas*, with a proportion of 19% (Fig. 3C).

## Blood cmDNA is enriched in disease-related information

Different microbial compositions are often associated with different functions. We first screened the characteristic microbiota in the samples from different groups using linear

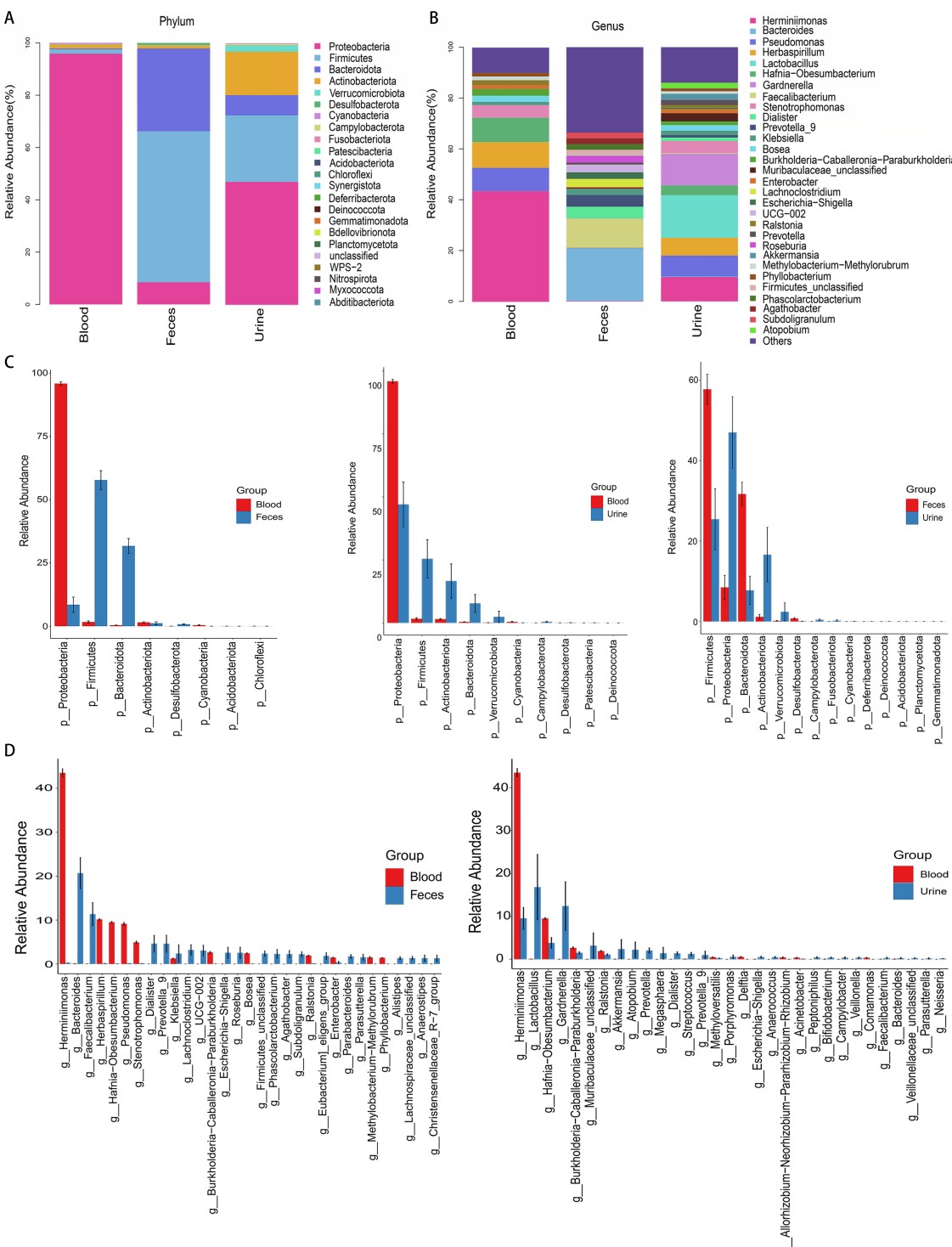

**FIG 2** Blood cmDNA indicates a unique species composition. (A) Stacked bar chart of species abundances at the phylum level. The horizontal axis of the plot represents the grouping of samples, and the vertical axis represents the relative abundance of a category; different colors correspond to different species at the same level. (B) Stacked bar chart of species abundances at the genus level. (C) The Mann–Whitney U test was used to analyze significant differences at the phylum level and showed that the blood cmDNA had a unique microbial composition. (D) The Mann–Whitney U test was used for significant difference analysis at the genus level between different groups.

discriminant analysis (LDA) effect size (LEfSe) analysis; Fig. 4A shows the significantly different species with LDA scores greater than 4.9. The results showed that the

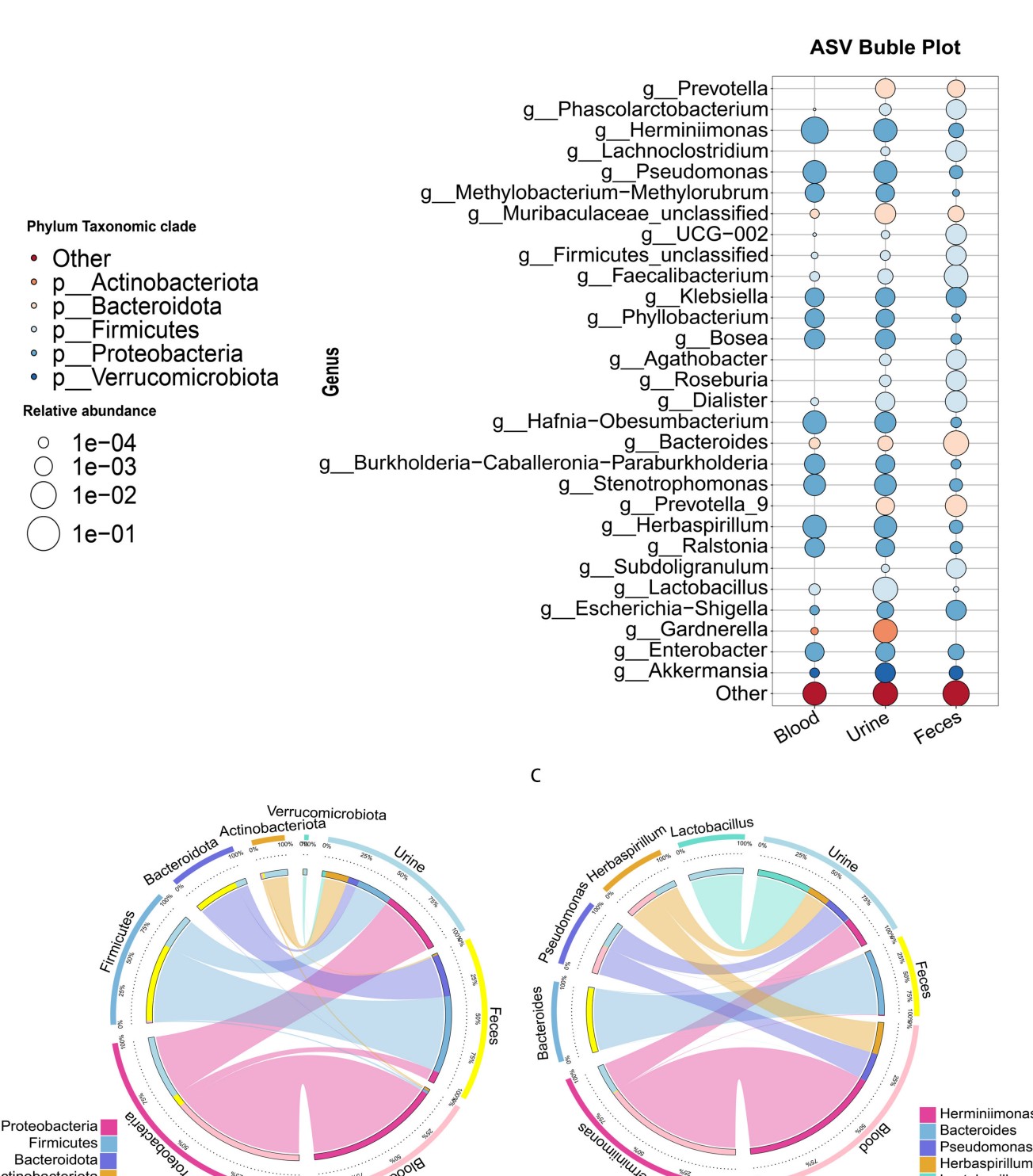

**FIG 3** Blood cmDNA indicates unique dominant species. (A) There were significant differences in dominant species between the different types of samples. The size of the circle represents the relative abundance at the genus level, while its color indicates the species at the phylum level in the corresponding taxonomic category. (B) The left half shows the top five phyla in terms of abundance along with the corresponding abundance information, and the right half shows the grouping information, with greater widths indicating greater abundance. (C) Distribution status of the top five genera in terms of abundance.

characteristic species with differences were significantly different between the samples from the three groups. The urine group was dominated by *o__Lactobacillales* and

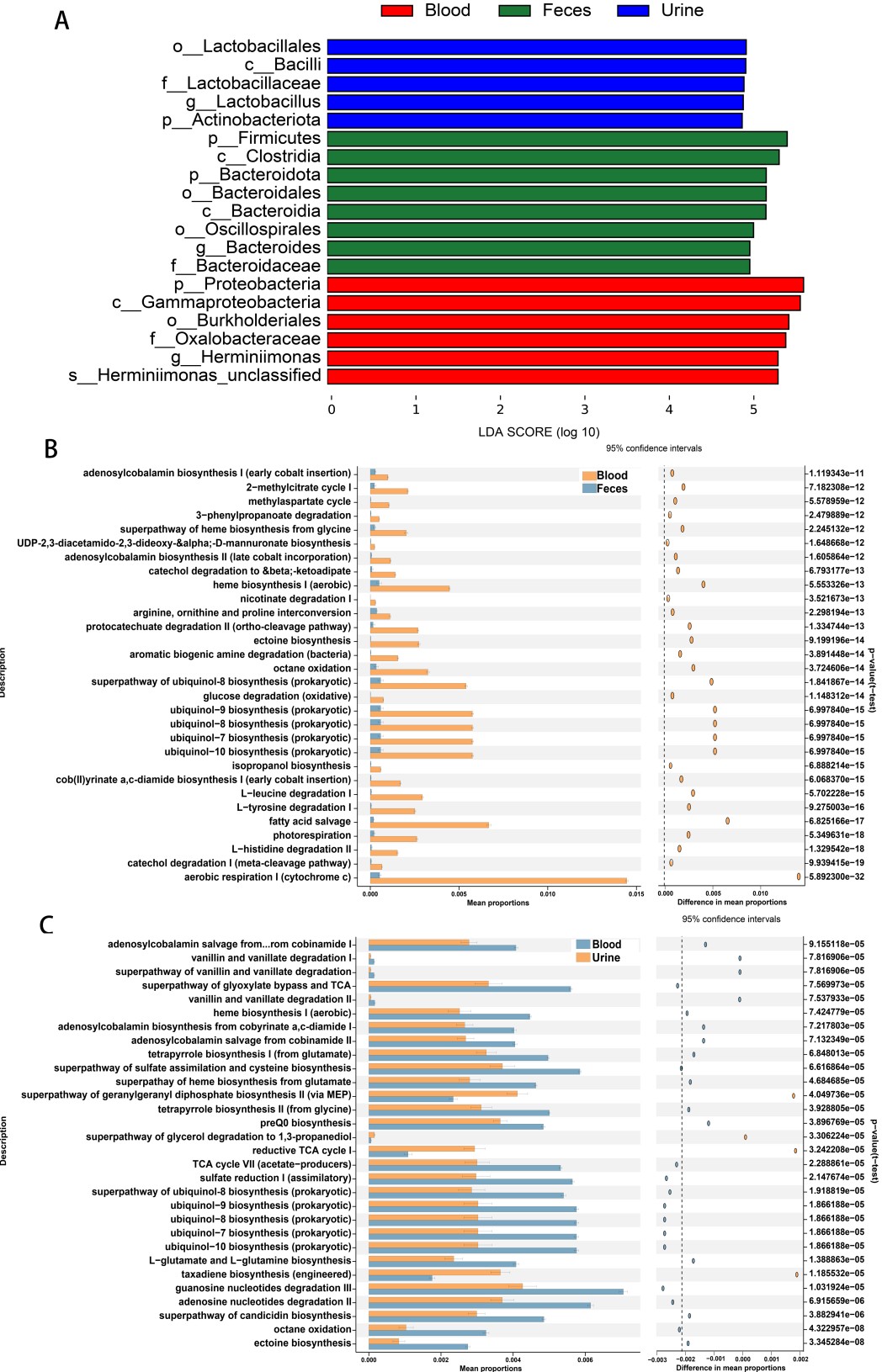

FIG 4  Blood cmDNA indicates unique signature microbiota and functions. (A) Bar plot of the distributions of differential species across samples from different groups. (B) The top 30 associated features with *P* values < 0.05 in blood and fecal samples were compared with 95% confidence intervals. (C) The top 30 associated features with *P* values < 0.05 in blood and urine samples were compared with 95% confidence intervals.

c__Bacilli, which were the most abundant species overall. In the fecal group, the highest abundances were found for p__Firmicutes and c__Clostridia. On the other hand, in the blood group, the most abundant species were p_Proteobacteria and c_Gammaproteobacteria. Subsequently, using the MetaCyc database, we employed PICRUSt2 analysis to explore the functions present within diverse sample groups. The importance of cmDNA in blood in maintaining the normal physiological functions of cells surpasses that of the fecal microbiota. This process involves multiple critical metabolic pathways, including ubiquinol-7 to 10 biosynthesis, fatty acid salvage, and aerobic respiration I. These pathways are crucial in cellular energy metabolism, antioxidant mechanisms, and cell signaling. More impressively, blood samples exhibited stronger associations with these pathways (Fig. 4B). The functions of urine and blood cmDNA are similar, as they both involve energy metabolism and synthetic processes. These include adenosine nucleotide degradation, the tricarboxylic acid cycle (TCA) cycle, and guanosine nucleotide degradation. However, it seems that blood samples exhibit a greater degree of enrichment in these pathways than urine samples (Fig. 4C).

## Blood cmDNA exhibits distinctive bacterial phenotypes

Drawing from our results in Fig. 4, we observed that peripheral blood cmDNA is intricately linked to diverse metabolic processes and exhibits marked disparities when compared to fecal samples. To further investigate whether these distinctions are associated with the microbiome phenotype of peripheral blood cmDNA, we investigated nine significant microbial phenotypes, namely, aerobic (Fig. 5A), anaerobic (Fig. 5B), mobile element-containing (Fig. 5C), facultatively anaerobic (Fig. 5D), biofilm-forming (Fig. 5E), gram-negative (Fig. 5F), gram-positive (Fig. 5G), potentially pathogenic (Fig. 5H), and stress-tolerant (Fig. 5I) microbes. Detailed comparative information for each group of samples with different phenotypes is provided in Table S3.

The microbial phenotypic characteristics observed in the blood samples indicated greater concentrations of aerobic bacteria, mobile element-containing bacteria, facultatively anaerobic bacteria, biofilm-forming bacteria, and gram-negative bacteria than those observed in the stool samples. Moreover, the cmDNA present in the blood samples was predominantly derived from gram-negative microorganisms, while the cmDNA in the fecal samples predominantly contained gram-positive microbes. In addition, compared to the microbial DNA in fecal and urine samples, the cmDNA present in blood samples presented a greater relative abundance of potentially pathogenic and stress-tolerant microorganisms. Similarly, the urine samples also exhibited results analogous to those obtained from the blood samples, but the relative abundance in the urine samples was lower than that observed in the blood samples.

## Gut microbes are not a major source of blood cmDNA

We also performed preliminary detection of the source of cmDNA in peripheral blood by utilizing Source Tracker (v 1.0). The intestine, the largest digestive organ in the human body, is the main site of microbial colonization. However, only 7.167% of the cmDNA in the peripheral blood samples was similar to the microbial DNA found in the fecal samples (Fig. 6A). Moreover, the main source of cmDNA in urine was cmDNA from peripheral blood (Fig. 6B).

## DISCUSSION

Due to its potential significance for normal physiology and immune function in healthy individuals, the importance of cmDNA in blood is gaining recognition (11). The source of blood cmDNA, whether it is from gut microbes or other sources, remains unclear, and emerging evidence suggests that cmDNA in the blood may originate from translocating microbes (3, 13). We explored the unique diversity and composition of blood cmDNA, comparing it to gut microbial DNA from the same group of volunteers. These findings

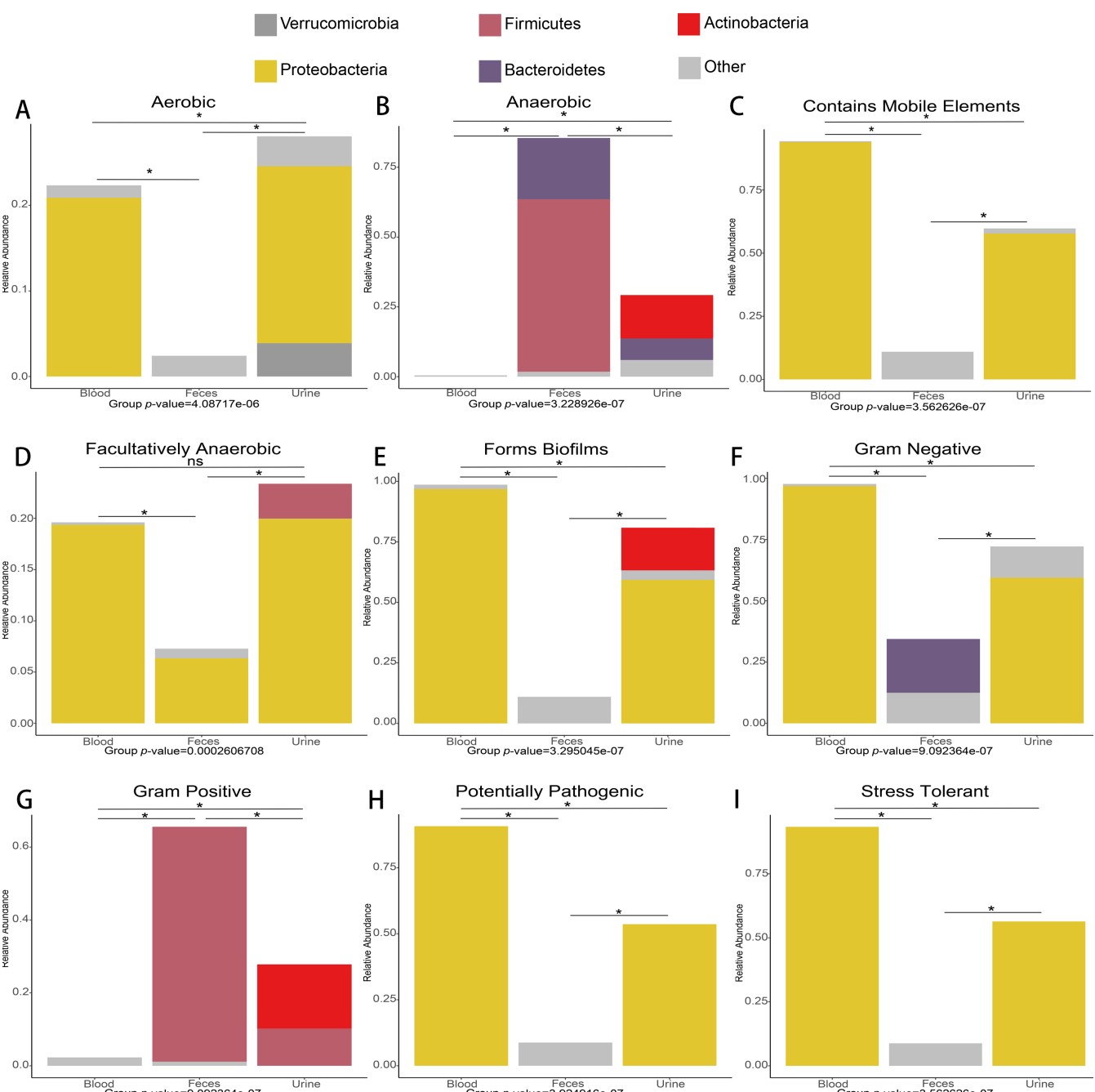

**FIG 5** Blood cmDNA is associated with unique bacterial phenotypes. The relative abundance profiles of species exhibiting different phenotypes in different groupings are illustrated as columnar stacking plots. The abscissa in the figure shows the grouping name, the ordinate shows the relative abundance of species predicted to have different phenotypes at the phylum level, different species can be distinguished based on color, and the columns (high and low) distinguish the relative abundance of different species in different groupings. Nine major potential phenotypes were detected, namely, aerobic (A), anaerobic (B), mobile element-containing (C), facultative anaerobic (D), biofilm-forming (E), gram-negative (F), gram-positive (G), potentially pathogenic (H), and stress-tolerant (I) microbes. The Kruskal–Wallis test was used to assess differences between groups, and the pairwise Mann–Whitney-Wilcoxon test was used to identify groups with differences. * indicates a significant difference compared to the blood group.

suggest that the origins and roles of cmDNA may be far more wide-ranging and complex than previously anticipated.

The debate around the origin of blood cmDNA is ongoing, with a previous consensus on a unique core blood microbiome (7, 15, 16). However, Tan et al. study challenges this

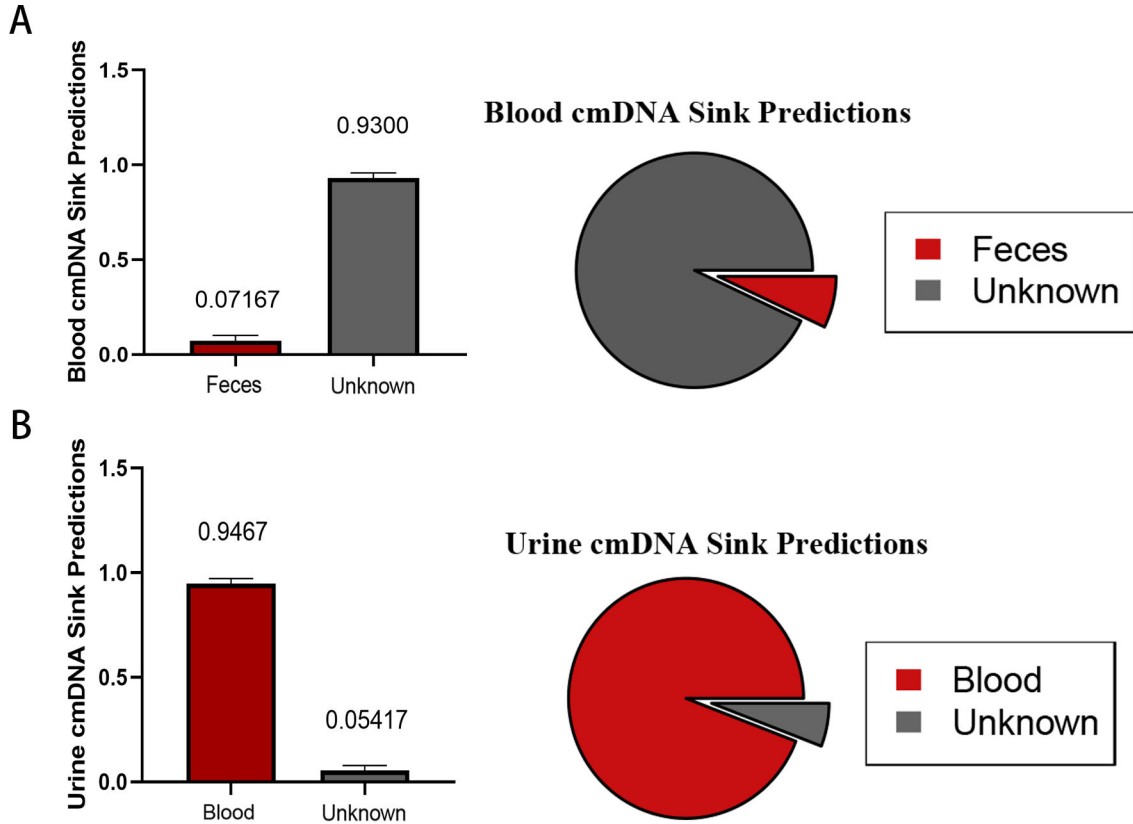

**FIG 6** Identification of the source of cmDNA in peripheral blood and urine. (A) The different sections in the pie charts represent the proportions of different microbial origins. The larger the area, the greater the proportion represented. The term "unknown" represents an unknown classification of origin. The mean proportion of blood cmDNA derived from the intestine was 7.167%. (B) Peripheral blood cmDNA was the main source of cmDNA in urine.

phenomenon by indicating the absence of a unique blood microbiome and suggesting that most microbial DNA in healthy blood is symbiotic and may enter during collection (13). We agree that there is no core blood microbiome, but we argue against the idea that microbial DNA is acquired during blood draw. Because in our research, no viable microorganisms were detected in the blood. We propose that microbial DNA in blood mainly originates from microbial translocation, including natural movement within the body and movement due to injuries, infections, or compromised barriers. Healthy blood likely contains more microbial genetic material than live bacteria, as most healthy donors show no bacterial growth (17). Research indicating that microorganisms can migrate from various body parts to tumors via the bloodstream supports this idea of translocation (18). Hence, microbial translocation, rather than intrinsic presence, is the predominant reason for cmDNA in peripheral blood (3).

We theorized that microbes in healthy individuals come from translocations, particularly from the gut, leading us to compare fecal and blood samples from the same volunteers to avoid individual differences. Our results revealed significantly greater microbial diversity in fecal and urine samples than in blood samples, with approximately 1,600 species in fecal and urine vs 600 species in blood, supporting previous findings of substantial diversity differences (19). Our study revealed distinct microbial compositions in blood, fecal, and urine samples. Blood and urine were predominantly populated by Proteobacteria (95% in blood and 45% in urine), in contrast to fecal samples, where Firmicutes and Bacteroidota made up over 90% of the microbiome. These findings highlight substantial differences between the gut microbiota and the cmDNA in peripheral blood (16, 20). Further analysis indicated that blood and urine share similar dominant microbes, diverging significantly from those in fecal samples. This raises

an unresolved question: if gut microbes are translocated to the blood, why is there such a disparity in microbial diversity between the sample types?

We examined the microbial composition in blood, fecal, and urine samples and found marked similarities between microbial structures in blood and urine but significant differences compared to fecal samples. Blood cmDNA was enriched in functions crucial for biochemical processes such as synthesis, energy conversion, and toxin degradation. Notable functional disparities, such as in ubiquinol biosynthesis and aerobic respiration, were observed between blood and fecal samples, highlighting distinct survival and adaptability mechanisms (21). The similarity in cmDNA functions between blood and urine samples might reflect common origins or release processes. Recent research into blood cmDNA has elucidated the peculiar presence of microbial DNA in the bloodstream, providing initial insights into microbial DNA circulation within the human body.

Based on the above research, we discovered a unique microbial composition and function hidden in blood cmDNA. Compared with those in the fecal samples, significant differences in the blood samples were observed, which led us to consider the relationship between blood cmDNA and gut microbial DNA. We hypothesized that cmDNA in the blood may be acquired through a selective mechanism.

Microbial phenotype analysis corroborated our hypothesis, revealing distinct cmDNA profiles between blood and fecal samples. Blood had fewer anaerobes and more aerobes and facultative anaerobes than the anaerobic-rich fecal microbiota. Predominantly, gram-negative bacteria were found in the blood, in contrast with fecal gram-positive bacteria. Blood cmDNA also displayed traits associated with mobility, biofilm formation, pathogenicity, and stress response, which were nearly absent in fecal samples. This finding suggested blood cmDNA selection rather than direct translocation from the gut.

Our traceability analysis indicated that only approximately 7% of the blood cmDNA was similar to fecal microbial DNA, with a near absence of gram-positive and anaerobic bacteria in peripheral blood, suggesting that further research is needed to determine the specific processes involved. The prevailing hypothesis on microbial or cmDNA translocation to the bloodstream involves increased intestinal permeability (22). A healthy gut microbiota is critical for maintaining the gut barrier, including tight junction integrity and immune function. The hypothesis linking heart failure to intestinal leakage suggests that heart failure-induced edema and barrier dysfunction may enable the gut microbiota to enter the circulatory system, causing endotoxemia and aggravating inflammation (23). However, microbial translocation due to increased permeability is considered random, which contrasts with our findings of selective migration and poses significant health risks, especially in pathological states (24). Another possible explanation involves the differential selective pressures exerted by aerobic and anaerobic environments on blood cmDNA and the fecal microbiota. However, importantly, such environmental selection primarily affects living microorganisms (25). As both our research findings and recent reports have revealed, there are virtually no live microorganisms in the blood; instead, they exist in the form of cmDNA (11, 13, 26). However, there is little conclusive evidence that an oxygen environment or differences in nutrient elements can cause drastic changes in microbial phenotypes, such as from gram negative to gram positive.

We suggest that extracellular vesicles, or membrane vesicles (MVs) from microorganisms, are the selective mechanism for cmDNA entry into the bloodstream (27). These MVs, which are 20–400 nm in size, are involved in the transfer of DNA, proteins, and other molecules and are implicated in immunomodulation, therapeutic applications, and intercellular communication (28, 29). They facilitate the transport of microbial genetic material, enabling bacteria to convey hydrophobic compounds to host cells without eliciting an immune response. MVs differ in composition depending on their microbial origin, which dictates their functionality and genetic material delivery (30). Intestinal MVs, which are absorbed by epithelial cells and capable of reaching distant organs through endothelial barriers, reflect the nature of the microbial community (31). Our research suggested that blood cmDNA is predominantly derived from gram-negative bacteria, in contrast to fecal cmDNA, which is more representative of gram-positive

bacteria. This could be related to differences in MVs from these bacterial groups, with gram-negative MVs known for their efficient internalization mechanisms and gram-positive MVs entering cells through slower pathways, such as clathrin-mediated endocytosis (32, 33). Notably, surface components such as lipopolysaccharide on gram-negative MVs could affect MVs uptake by host cells, indicating that surface properties are pivotal in determining the selective passage of MVs into the bloodstream (31).

Our initial data revealed patterns of nonrandom cmDNA presence in the peripheral blood of healthy subjects, distinct from the patterns observed in gut microbial DNA profiles. This observation suggested a potential selective mechanism that influences the phenotype and function of cmDNA. Although our findings offer valuable insights into the origins of cmDNA within peripheral blood, while these insights provide a starting point for understanding the presence of cmDNA in peripheral blood, the limitations of our study—including its relatively small cohort of 26 participants—must be acknowledged. This limitation calls for a cautious approach to interpreting our results and highlights the need for further investigation with more expansive and diverse participant groups to substantiate our initial observations. Moreover, we were restricted to healthy volunteers, which limits our generalizability across various demographic conditions, including individuals with differing health statuses, ethnic backgrounds, and dietary habits. Thus, our results may not be generalizable to the broader population. Additionally, our hypothesis regarding the selective mechanisms governing cmDNA in the bloodstream remains conjectural and is not supported by definitive causal evidence. To establish a clear causal linkage, more rigorous experimental research is essential.

## MATERIALS AND METHODS

### Sample collection and preparation

This study was approved by the institutional review board and ethics committee of the First Hospital, Jilin University (2023–529) and adhered to the ethical standards of the Declaration of Helsinki; informed consent was obtained from all participants. A total of 36 healthy volunteers were enrolled; these volunteers did not have any chronic illnesses and did not receive any medication treatment in 3 months prior to inclusion in the study. Sterile lancets were used to collect blood, midstream urine, and fecal samples. Measures were taken to prevent microbial contamination from the surfaces of the sampling sites of the volunteers. Specifically, we used sterile swabs to collect microbial samples from the surface of the blood collection site, urethral orifice, and anus of the volunteers; these samples were treated as negative controls, and the microbial data from these areas were excluded from the subsequent experimental analysis. The blood and urine samples were subjected to centrifugation at 3,000 rpm for 15 minutes. The resulting supernatants from urine, plasma, and feces were subsequently stored at −80°C for further processing and analysis.

### Blood culture

Blood culture was conducted using the BACTEC 9240 system (Becton, Dickinson and Company, New Jersey, USA). Due to the limited volume of the plasma samples, we selected BACTEC PDES Plus culture vials, designed for pediatric or low-volume samples, for cultivation. Following the manufacturer's recommendations, we added 0.5 mL of the plasma sample to each culture vial. The default incubation period was set at 7 days. If bacteria grow in a culture bottle, they can interact with the fluorescent substances within it, leading to the excitation of fluorescence. The instrument then detects this fluorescence, utilizing it as a proxy for assessing bacterial growth. If no bacterial growth was detected within this timeframe, the result was considered negative.

## DNA extraction

A SanPrep Column DNA Gel Extraction Kit was obtained from Sangon Biotech (Shanghai, China). Following the manufacturer's instructions, we used the cetyltrimethylammonium bromide (CTAB) method to extract DNA from various samples, including blood, urine, fecal, and negative controls, while nuclease-free water was used as a blank control to ensure experimental accuracy (34). We adhered to the same rigorous protocols for processing all the samples, negative controls, and blank controls. To comprehensively consider potential sources of contamination, we included a range of control groups, such as sampling controls, DNA extraction controls, and no-template PCR amplification controls, accounting for both the hospital and laboratory environments as well as various stages of sample handling and processing, as shown in Table S4. These measures were adopted to closely monitor and minimize potential contamination during the experiment. The total DNA was then eluted in 50 µL of elution buffer and stored at −80℃ until further PCR amplification and 16S rDNA sequencing were conducted by LC-Bio Technology Co., Ltd., in Hangzhou, Zhejiang Province, China.

## PCR amplification and 16S rDNA sequencing

The primers used in the study were universal sequencing primers with specific barcodes for each sample. The forward primer 341F (CCTACGGGNGGCWGCAG) and reverse primer 805R (GACTACHVGGGTATCTAATCC) were PCR-purified bacterial 16S rRNA gene amplicon PCR primers (35). PCR amplification was carried out in a mixture with a total volume of 25 µL containing 25 ng of template DNA, 12.5 µL of PCR Premix Ex Taq (Tokyo, Japan), 2.5 µL of each primer, and PCR-grade water to adjust the volume. The prokaryotic 16S fragments were amplified with the following PCR protocol: initial denaturation at 98℃ for 30 seconds; 32 cycles of denaturation at 98℃ for 10 seconds, annealing at 54℃ for 30 seconds; and extension at 72℃ for 45 seconds. A final extension was performed at 72℃ for 10 minutes. The PCR products were verified through 2% agarose gel electrophoresis. Ultrapure water, instead of a sample solution, was used as a negative control throughout the DNA extraction process to exclude the possibility of false-positive PCR results. The PCR products were purified using AMPure XT beads (Beckman Coulter Genomics, Danvers, MA, USA) and quantified using Qubit (Invitrogen, USA). The amplicon pools were prepared for sequencing, and the size and quantity of the amplicon library were assessed using an Agilent 2100 Bioanalyzer (Agilent, USA) and the Library Quantification Kit for Illumina (Kapa Biosciences, Woburn, MA, USA), respectively. Finally, the libraries were sequenced on a NovaSeq PE250 platform. The raw Illumina sequencing data have been archived as Bioproject (ID: PRJNA1042891).

## Sample exclusion procedure

We first performed sequencing on 36 plasma samples. Subsequently, we utilized a similar method to exclude contaminant information, as described by Nejman et al. (34). After rigorous quality trimming and filtering, we removed low-complexity sequences of unknown taxonomic origin and human reads from the blood sample data. During the initial screening process, microbial DNA information inherent in the blood could not be detected in the samples from six patients after excluding DNA information from human sources and blank controls. Additionally, only a minimal amount of cmDNA information (<100 read pairs) remained after the same process in the samples from four patients, rendering it impossible to further exclude microbial information detected in the negative control samples. Ultimately, only the fecal and urine samples from the remaining 26 volunteers were included in the subsequent analysis.

## Data analysis

The samples were sequenced on an Illumina NovaSeq platform following LC-Bio's recommendations. Paired-end reads were assigned to each sample based on the unique

barcodes, and the barcode and primer sequences were subsequently removed. The fast length adjustment of short reads method was used to merge the paired-end reads (36). Fqtrim (v0.94) was used to filter the raw reads and obtain high-quality clean tags. Vsearch (v2.3.4) was used to remove chimeric sequences (37). After dereplication with divisive amplicon denoising algorithm 2 (DADA2), a feature table and sequence were generated (38). Alpha and beta diversity values were calculated and normalized to assess species diversity and compare samples. The SILVA (release 138, https://www.arb-silva.de/documentation/release-138/) classifier was used for feature abundance normalization. Five indices were calculated for alpha diversity analysis using QIIME2's classify-sklearn taxonomy classifier (39). Beta diversity was calculated by QIIME2, and graphs were generated using an R package. ANOSIMs, which are based on unweighted unifrac, were used to assess the statistical significance of differences between original sample groups through permutation tests. The Basic Local Alignment Search Tool was used to align the sequences, and representative sequences were annotated with the SILVA database (40). Stage-dependent features were identified via LEfSe with default settings (e.g., LDA score >2). Microbial functions were analyzed via phylogenetic investigation of communities through reconstruction of unobserved states (PICRUSt2) based on ASVs from 16S rDNA sequencing data, predicting metabolic functions from the MetaCyc database (https://github.com/picrust/picrust2) (41). Differences in predictions were analyzed using Welch's $t$ test with Benjamini−Hochberg FDR correction in STAMP (42). Phenotypic analysis of blood or urine cmDNA and fecal microorganisms was conducted utilizing the reported BugBase method (available at https://bugbase.cs.umn.edu/upload.html) (43). The relative abundance of distinct phenotypes across various species within distinct groups is presented through a stacked bar chart. Species tracing analysis was performed with SourceTracker (v1.0) according to previously reported methods (44).

## Statistical analysis

After obtaining statistics on species abundances, we utilized the Kruskal−Wallis test to compare samples between groups. If the results from the Kruskal−Wallis test were significant, we further conducted pairwise Mann−Whitney-Wilcoxon tests to identify groups with differences. The SD of each group was used to derive the results, and a $P$ value lower than 0.05 was considered statistically significant.

## ACKNOWLEDGMENTS

This manuscript is supported by the National Natural Science Foundation of China (82302591), the Health Commission of Jilin Province (2022JC059), and the Project of Jilin Provincial Department of Education (JJKH20241327KJ).

## AUTHOR AFFILIATIONS

[1]Department of Clinical Laboratory, The First Hospital of Jilin University, Changchun, China
[2]College of Medical Technology, Beihua University, Jilin, China

## AUTHOR ORCIDs

Taiyu Zhai ⓘ http://orcid.org/0000-0003-2843-7047
Jing Huang ⓘ http://orcid.org/0000-0003-1282-1546

## FUNDING

| Funder | Grant(s) | Author(s) |
| --- | --- | --- |
| Health Commission of Jilin Province (吉林省卫生健康委员会) | 2022JC059 | Jing Huang |

mSystems

| Funder | Grant(s) | Author(s) |
|---|---|---|
| MOST | National Natural Science Foundation of China (NSFC) | 82302591 | Taiyu Zhai |
| Project of Jilin Provincial Department of Education | JJKH20241327KJ | Wenbo Ren |

## AUTHOR CONTRIBUTIONS

Taiyu Zhai, Data curation, Formal analysis, Funding acquisition, Writing – original draft, Writing – review and editing | Wenbo Ren, Formal analysis, Software, Writing – review and editing | Xufeng Ji, Data curation, Software | Yifei Wang, Writing – review and editing | Haizhen Chen, Data curation | Yuting Jin, Software | Qiao Liang, Validation | Nan Zhang, Software | Jing Huang, Project administration, Resources

## DATA AVAILABILITY

The Strengthening the Organizing and Reporting of Microbiome Studies (STORMS) checklist (45) is available at https://doi.org/10.5281/zenodo.10602735.

## ADDITIONAL FILES

The following material is available online.

### Supplemental Material

**Table S1 (mSystems00008-24-s0001.xlsx).** Detailed comparison data of microorganisms at the phylum levels in the three groups.
**Table S2 (mSystems00008-24-s0002.xlsx).** Detailed comparison data of microorganisms at the genus levels in the three groups.
**Table S3 (mSystems00008-24-s0003.xlsx).** Detailed comparative information for each group of samples with different phenotypes.
**Table S4 (mSystems00008-24-s0004.docx).** Experimental and control group settings

### Open Peer Review

**PEER REVIEW HISTORY (review-history.pdf).** An accounting of the reviewer comments and feedback.

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
