## [Reviewer comments · mSystems]

Distinct compositions and functions of circulating microbial DNA in the peripheral blood compared to fecal microbial DNA in healthy individuals

Taiyu Zhai, Wenbo Ren, Xufeng Ji, Yifei Wang, Haizhen Chen, Yuting Jin, Qiao Liang, Nan Zhang, and Jing Huang

Corresponding Author(s): Jing Huang, The First Hospital of Jilin University

Review Timeline:

Submission Date:	January 3, 2024
Editorial Decision:	January 10, 2024
Revision Received:	February 6, 2024
Accepted:	February 9, 2024

Editor: James Brown

Reviewer(s): The reviewers have opted to remain anonymous.

Transaction Report:

DOI: <https://doi.org/10.1128/msystems.00008-24>

Re: mSystems00008-24 (Distinct composition and functions of circulating microbial DNA in the peripheral blood compared to fecal microbial DNA in healthy individuals)

Dear Prof. Jing Huang:

Please address my Editorial comments as detailed in the attached document.

Revision Guidelines

Sincerely,
James Brown
Editor
mSystems

Dear Editor,

We truly appreciate your feedback on our article. We have taken your comments into serious consideration and made the necessary changes to the text as advised. Your guidance has been instrumental in refining our manuscript, and we are grateful for your meticulous attention to detail and dedication to upholding the highest standards of academic publishing. Thank you for your invaluable input, and we look forward to hearing from you soon.

We have highlighted all the changes in the marked-up file. Moreover, to prevent potential grammatical or other errors resulting from significant revisions, the amended manuscript has already undergone a subsequent review by AJE (American Journal Experts) for enhancement, ensuring the document's quality. The following is a point-by-point response to your comments.

Comment 1: Page 2. Lines 54-56. Importance: Please delete last sentence as it is redundant: “These contributions underscore the significance of our study in expanding the current state of knowledge on cmDNA and its implications for human health.”

Response 1: We have made the requested change to the manuscript per your feedback. Thank you for your input.

Comment 2: Page 6. Lines 147-16 and throughout the manuscript. Add summary stats (test name and FDR-adj. P value) to statements of significance and non-significance. For example, lines 147: “The α -diversity analysis revealed significantly lower (Kruskal–Wallis test, FDR-adj. P values ≤ 0.05) Chao1 and Shannon indices (Figure 1B, 1C) and a significantly higher (Kruskal-Wallis test, FDRadj. P-values ≤ 0.05) Good’s coverage index (Figure 1D) in the blood samples than in the urine and fecal samples.

Response 2: Based on this comment, we have added statistical information to the appropriate places in the article, including results 1 and 2. (Page 6, Line 144-149. Page 7, Line 179-183.)

Comment 3: Page 6. Lines 159-161. Need to document ANOSIM analysis in Methods. Also, by stating R- values and P- values as well as sample size (N) in the text in parentheses (as mentioned above in point 2) you can eliminate Table 2. Also please expand any abbreviations at their first usage in the manuscript (i.e., Analysis of Similarity (ANOSIM)) as well as provide a citation for that analysis tool or database.

Response 3: We appreciate your valuable feedback. Regarding your comment on the need for additional information on the ANOSIM analysis in the Methods section, we have ensured that the relevant details are now included. To enhance clarity, we have also included the R values, P values, and sample sizes (N) in parentheses within the text in the Results section, thus eliminating the need for Table 2. Furthermore, we have expanded upon the abbreviations at their first usage in the manuscript, specifically in the "Data analysis" section of the Methods section. (Page 6, Line 157-159. Page 16, Line 450, 452-454, 459-470.)

Comment 4: Lines 176-181. I agree with Comment #6 made by Reviewer 2 that you need provide a statistical test for ASV comparisons in Figures 2C and 2D. FDR adj. P-values needed to be reported. I would also suggest including these values in a Supplemental Data file in MSEXcel format along with taxon names, mean values, standard deviations.

Response 4: Thank you for your feedback. We agree that a statistical test for ASV comparisons in Figures 2C and 2D is necessary, and we have provided FDR adj. P values in the revised manuscript. Additionally, we will include these values in a Supplemental Data file in Microsoft Excel. We appreciate your suggestions and will make the necessary changes to improve the quality of our work. (Page 7, Line 179-183.)

Comment 5: For Figure 5A-I, also mentioned by the reviewer, you have provided the whole group Pvalue but not the pairwise FDR-adj. values between each sample type, Blood, Feces and Urine. These could be added as horizontal lines above the tested pairs with "*" notations for significance level and "ns" for non-significance. I would

also suggest sharing the data used to generate this figure in another Supplementary data file. Further comments about Figure 5 and the phenotype analysis follow below. In general, it is good practice to include supplementary files with the data used to generate any figures so that others can replicate your analyses.

Response 5: Thank you for your insightful feedback. During our earlier data processing stage, we generated the raw statistical data utilized in creating these figures. In accordance with your suggestion, we have compiled the underlying data for all nine figures in a supplementary file for easier accessibility and comprehensive presentation of our findings. We believe this approach will enhance the clarity and reproducibility of our research. We appreciate your guidance and will continue to strive for excellence in our work. (Figure 5. Page 9, Line 232-233.)

Comment 6: Line 209. Expand “KO” abbreviation and add citation to PICRUSt2.

Response 6: Based on your feedback, we have added references to the PICRUSt2 analysis in Line 464 and revised the KO database to the MetaCyc database. (Page 8, Line 211. Page 17, Line 468.)

Comment 7: Line 208-221. Figure 4B. I suggest caution in emphasizing the linkage to human pathways and diseases which is likely a spurious effect of using the KEGG database. KEGG generalizes enzyme functions across eukaryotes and prokaryotes. KEGG updates are also restricted to paid subscribers. PICRUSt2 now uses MetaCyc as its default database, which is prokaryote specific, open source and regularly updated. It might be worthwhile redoing your analyses with MetaCyc.

Response 7: Thank you for your detailed feedback on Figure 4B. Your comments regarding the KEGG database are very pertinent, and we recognize the potential issues associated with using KEGG. Based on your suggestions, we have completed the reanalysis using MetaCyc as the default database to ensure the accuracy and relevance of the results. Furthermore, we have modified the descriptive content related to Figure 4B, including the Abstract, Results and Discussion sections. (Page 1, Line 21-23. Page 2, Line 44. Page 8, Line 212-222. Page 11, Line 290-300.)

Comment 8: Lines 225-226. Please add a citation to your previous research.

Response 8: We apologize for the misunderstanding caused by our mistake. The "previous study" referred to the results in Figure 4, not the previously published study. Given that we have reanalyzed the results in Figure 4B and 4C, we have also made corresponding changes to the descriptive content here. (Page 8, Line 224-227.)

Comment 9: Line 230-231 and Figure 5. The proper spelling and leger case is, "gram positive" and "gram negative" without a hyphen. Please change throughout.

Response 9: Thank you for your feedback. We have made the necessary changes to "gram positive" and "gram negative" throughout the text, removing the hyphens as suggested. Other relevant sections have also been updated accordingly.

Comment 10: Line 226-232 and Figure 5. You need to include a Supplemental Table of the specific bacterial genera/species which compose each phenotype. Also provide the references to your phenotype paper. Also include in the Methods how phenotypes were assigned.

Response 10: Thank you for your feedback. We will include a Supplemental Table in the revised manuscript, which lists the specific bacterial genera/species that compose each phenotype. We will also provide references for our phenotype paper and include detailed information on how phenotypes were assigned in the Methods section. (Page 9, Line 232-233. Page 17, Line 471-475.)

Comment 11: Line 246. Add the reference to Source Tracker.

Response 11: Thank you very much for your valuable feedback. In the previous version of the manuscript, we comprehensively cited the relevant references regarding Source Tracker in the "Data analysis" section; hence, no redundant references were added to the "Results" section of Line 246. We highly value your feedback and are always looking for ways to improve the quality of our manuscripts. If you have any additional comments or suggestions or if we have misinterpreted your feedback, please feel free to let us know. (Page 17, Line 476.)

Comment 12: Line 253. This statement is highly speculative. Please change to: "... due to its potential significance..."

Response 12: We have made the requested change. Additionally, references have been added to the relevant content. Thank you for your feedback. (Page 9, Line 253.)

Comment 13: Line 254-257. Please add citations to: "... evidence suggests that cmDNA in the blood may originate from translocating microbes."

Response 13: We have made the requested change. Thank you for your feedback. (Page 9, Line 257.)

Comment 14: Lines 335-347. There is no causal evidence from your study linking cmDNA to human diseases. Furthermore, the fact that these are health subjects and no diseased cohorts were sampled suggests that you are over interpreting the results from PICRUSt2.

Response 14: This is indeed a very important issue. According to comment 7, we have reanalyzed this part and made corresponding modifications to the relevant content. (Page 11, Line 290-300.)

Comment 15: Lines 352-356. The most plausible explanation for species cmDNA differences in the blood and stool is likely due to selection driven the anaerobic versus aerobic environment.

Response 15: Thank you for your insightful feedback on our research. Your observation is indeed an important consideration. It is critical to clarify that cmDNA in the blood does not indicate the presence of live microorganisms; instead, it comprises DNA fragments from microorganisms that are present in the bloodstream. These fragments can originate from lysed live microbes or apoptotic cells or can be directly released by certain bacteria as a part of their life cycle.

The microbial profile revealed through blood cmDNA analysis does not necessarily mirror the living microbial community one might predict based on environmental

selection pressures alone; it represents the DNA from microbes that have managed to enter the bloodstream. The prevalence of aerobic, mobile element-containing, facultatively anaerobic, biofilm-forming, gram negative, potentially pathogenic, and stress-tolerant phenotypes in blood cmDNA might result from various factors, including the translocation of microbes from distinct body sites into the bloodstream or the selective capture of microbial DNA by extracellular vesicles.

Conversely, the microbial communities in stool samples are shaped by the gut's anaerobic conditions, which predominantly support anaerobic and gram positive microbes. Consequently, the microbial profiles in stool are more directly influenced by the internal environment of the gastrointestinal tract.

Given these considerations, the differences in microbial phenotypes between blood cmDNA and stool microbiota can be ascribed to a combination of factors. These include the different selection pressures imposed by the respective environments, the mechanisms of microbial translocation to the bloodstream, and the preferential packaging of DNA from various microbial phenotypes into extracellular vesicles.

We appreciate your comment, as they have encouraged a more thorough contemplation of these intricate interactions and the inclusion of a comprehensive discussion in our manuscript. We believe that this work will enhance both our understanding and that of the broader research community regarding the complexities of cmDNA distribution in diverse biological samples. We are eager to receive additional valuable insights. (Page 12, Line 326-333.)

Comment 16: The Discussion is very long and should be reduced, particularly with the speculative notes on disease linkages and extracellular vesicles for which your study was not designed to address. The first few paragraphs of the Discussion can be greatly reduced. You need to focus on the core findings and comparisons to previous published findings. Please add a paragraph to discuss the limitations of your study (i.e., no disease patients, lack of bacterial viability data, etc.).

Response 16: We have streamlined the Discussion section according to your feedback,

focusing more on the core findings and comparisons to previously published research. We have also included a paragraph discussing the limitations of our study, such as the absence of disease patients. Thank you for your valuable input, which has helped us improve the quality of our research.

Comment 17: Data availability. The provided url is not active. You should deposit these data in National Centre for Biotechnology Information Sequence Read Archive (SRA) under a proper BioProject ID. The one provided in your manuscript (PRJNA1042891) is not active.

Response 17: Since we have already started follow-up research, for the purpose of data protection, the BioProject ID (PRJNA1042891) we provided before has set an open time. According to your advice, we have now fully opened this part of the data.

Comment 18: Computer code availability. Please deposit any customized code in GitHub or similar public repository.

Response 18: In this study, we did not involve any customized code. All the codes are from already open-access repositories, and the corresponding citation information has been added to the text.

Re: mSystems00008-24R1 (Distinct compositions and functions of circulating microbial DNA in the peripheral blood compared to fecal microbial DNA in healthy individuals)

Dear Prof. Jing Huang:

Your manuscript has been accepted, and I am forwarding it to the ASM production staff for publication. Your paper will first be checked to make sure all elements meet the technical requirements. ASM staff will contact you if anything needs to be revised before copyediting and production can begin. Otherwise, you will be notified when your proofs are ready to be viewed.

Cover Image Submissions: If you would like to submit a potential Featured Image, please email a file and a short legend to msystems@asmusa.org. Please note that we can only consider images that (i) the authors created or own and (ii) have not been previously published. By submitting, you agree that the image can be used under the same terms as the published article. Image File requirements: TIF/EPS, 7.5 inches wide by 8.25 inches tall (at least 2,250 pixels wide by 2,475 pixels tall), minimum 300 dpi resolution (600 dpi preferred), RGB, and no figure elements, e.g., arrows or panel labels. The legend should be a short description of the image, 1-2 sentences recommended.

We recognize that the video files can become quite large, so to avoid quality loss ASM suggests sending the video file via <https://www.wetransfer.com/>. When you have a final version of the video and the still ready to share, please send it to mSystems staff at msystems@asmusa.org.

Sincerely,
James Brown